# Gold-standard solutions to the Schrödinger equation using deep learning: How much physics do we need?

Leon Gerard[†,a,*], Michael Scherbela[†,*], Philipp Marquetand[†,§], and Philipp Grohs[†,‡,¶]

[†]Research Network Data Science @ Uni Vienna, Kolingasse 14-16, A-1090 Vienna, Austria
[‡]Faculty of Mathematics, University of Vienna, Oskar-Morgenstern-Platz 1, A-1090 Vienna, Austria
[§]Institute of Theoretical Chemistry, Faculty of Chemistry, University of Vienna, Währinger Straße 17, 1090 Vienna, Austria
[¶]Johann Radon Institute for Computational and Applied Mathematics, Austrian Academy of Sciences, Altenbergerstrasse 69, 4040 Linz, Austria
[a]leon.gerard@univie.ac.at
[*]These authors contributed equally

## Abstract

Finding accurate solutions to the Schrödinger equation is the key unsolved challenge of computational chemistry. Given its importance for the development of new chemical compounds, decades of research have been dedicated to this problem, but due to the large dimensionality even the best available methods do not yet reach the desired accuracy. Recently the combination of deep learning with Monte Carlo methods has emerged as a promising way to obtain highly accurate energies and moderate scaling of computational cost. In this paper we significantly contribute towards this goal by introducing a novel deep-learning architecture that achieves 40-70% lower energy error at 6x lower computational cost compared to previous approaches. Using our method we establish a new benchmark by calculating the most accurate variational ground state energies ever published for a number of different atoms and molecules. We systematically break down and measure our improvements, focusing in particular on the effect of increasing physical prior knowledge. We surprisingly find that increasing the prior knowledge given to the architecture can actually decrease accuracy.

## 1 Introduction

**The challenge of the Schrödinger Equation** Accurately predicting properties of molecules and materials is of utmost importance for many applications, including the development of new materials or pharmaceuticals. In principle, any property of any molecule can be calculated from its wavefunction, which is obtained by solving the Schrödinger equation. In practice, computing accurate wavefunctions and corresponding energies is computationally extremely difficult for two reasons: First, the wavefunction is a high-dimensional function, depending on all coordinates of all electrons, subjecting most methods to the curse of dimensionality. Second, the required level of accuracy is extremely high. While total energies of small molecules are typically hundreds of Hartrees, the chemically relevant energy differences are on the order of 1 milli-Hartree as depicted in Fig. 1. Decades of research have produced a plethora of methods, which all require a trade-off between accuracy and computational cost: On one end of the spectrum are approximate methods such as Hartree-Fock (HF) or Density Functional Theory (DFT), which was awarded the Nobel prize in 1998. These methods can treat thousands of particles but can often only crudely approximate chemical properties. On the other end of the spectrum are "gold-standard" methods such as FCI (Full Configuration Interaction) or CCSD(T) (Coupled Clusters Singles Doubles (Perturbative Triples))

which yield energies that often closely agree with experiments, but can only treat up to 100 particles. Despite all these efforts, even for small molecules there do currently not exist highly accurate energy calculations. A 2020 benchmark of state-of-the-art methods for the benzene molecule found a spread of 4 mHa across different methods [1] – as a matter of fact, our results show that the absolute energies calculated in [1] are off by at least 600 mHa, due to the small basis set used in their calculations. An important characteristic of a method is the type of approximation being made: Hartree-Fock or

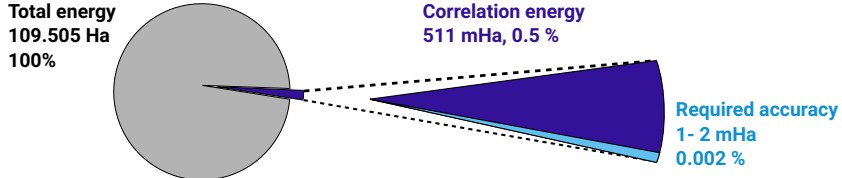

Figure 1: Conceptual visualization of the required accuracies on the example of an $N_2$ molecule: For chemical applications, total energies must achieve accuracies of $\sim 99.998\%$.

FCI are "variational", meaning that their predicted energies at least upper-bound the ground-truth energy. Since a lower energy is always guaranteed to be a closer approximation of the true energy, this makes assessment of these methods straight-forward. In contrast, CCSD(T) or DFT do not have any guarantees on the accuracy of their results. The approximations resulting from such methods, while working well for many common systems, often fail for chemically challenging situations such as breaking of chemical bonds [2, 3].

**Deep-learning-based variational Monte Carlo**   Combining deep learning and Monte Carlo methods has recently emerged as a promising new approach for solving the Schrödinger equation [4, 5]. These methods offer high accuracy, moderate scaling of computational cost with system size and obey the variational principle. Within a few years deep-learning-based methods have managed to outperform conventional high-accuracy methods for many different molecules, potentially defining a new gold-standard for high-accuracy solutions. In the Born-Oppenheimer approximation a molecule, consisting of $n_{\text{nuc}}$ nuclei and $n_{\text{el}}$ electrons, is fully described by its Hamiltonian in atomic units

$$H = -\frac{1}{2}\sum_i \nabla^2_{\boldsymbol{r}_i} + \sum_{i>j} \frac{1}{|\boldsymbol{r}_i - \boldsymbol{r}_j|} + \sum_{I>J} \frac{Z_I Z_J}{|\boldsymbol{R}_I - \boldsymbol{R}_J|} - \sum_{i,I} \frac{Z_I}{|\boldsymbol{r}_i - \boldsymbol{R}_I|}.$$

Here $\boldsymbol{R}_I$, $Z_I$, $I \in \{1, \ldots, n_{\text{nuc}}\}$ denote the coordinates and charges of the nuclei, $\boldsymbol{r} = (\boldsymbol{r}_1, \ldots, \boldsymbol{r}_{n_\uparrow}, \ldots, \boldsymbol{r}_{n_{\text{el}}}) \in \mathbb{R}^{3 \times n_{\text{el}}}$ denotes the set of $n_{\text{el}}$ Cartesian electron coordinates differentiated between $n_\uparrow$ spin-up and $n_\downarrow$ spin-down electrons. We define the inter-particle vectors $\boldsymbol{r}_{ij} = \boldsymbol{r}_i - \boldsymbol{r}_j$ and $\boldsymbol{\rho}_{iJ} = \boldsymbol{r}_i - \boldsymbol{R}_J$. All properties of the molecule depend on the wavefunction $\psi(\boldsymbol{r})$, which must fulfill the antisymmetry constraint: $\psi(\mathcal{P}\boldsymbol{r}) = -\psi(\boldsymbol{r})$ for any permutation $\mathcal{P}$ of two electrons with the same spin [6]. The wavefunction $\psi$ can be found as the solution to the Schrödinger equation $H\psi = E_0\psi$ with the ground-state energy and smallest eigenvalue $E_0$. By the Rayleigh-Ritz principle [7], the ground-state energy and the corresponding wavefunction can be found through minimization of the loss

$$\mathcal{L}(\psi_\theta) = \mathbb{E}_{\boldsymbol{r} \sim \psi_\theta^2(\boldsymbol{r})} \left[ \frac{H\psi_\theta(\boldsymbol{r})}{\psi_\theta(\boldsymbol{r})} \right] \geq E_0 \tag{1}$$

for a trial wavefunction $\psi_\theta$, parameterized by parameters $\theta$. The trial function $\psi_\theta$ is represented by a neural network and typically has the form

$$\psi_\theta(\boldsymbol{r}) = \sum_{d=1}^{n_{\text{det}}} \det \left[ \Lambda_{ki}^d(\boldsymbol{r}) \Omega_k^{d\alpha_i}(\boldsymbol{r}_i) \right]_{k,i=1,\ldots,n_{\text{el}}} \tag{2}$$

with $\Lambda_{ki}^d : \mathbb{R}^{3 \times n_{\text{el}}} \to \mathbb{R}$, $\Omega_k^d : \mathbb{R}^3 \to \mathbb{R}$, $\alpha_i \in \{\uparrow, \downarrow\}$, $i \in \{1, \ldots, n_{\text{el}}\}$, $k \in \{1, \ldots, n_{\text{el}}\}$. Each determinant is taken over a $n_{\text{el}} \times n_{\text{el}}$ matrix, with row-indices $k$ running over orbitals and column-indices $i$ running over electrons. The determinant enforces antisymmetry, $\Omega_k^d$ are envelope functions enforcing the boundary condition $\lim_{|\boldsymbol{r}| \to \infty} \psi_\theta(\boldsymbol{r}) = 0$, and $\Lambda_{ki}^d$ are neural networks. The local energy $\frac{H\psi}{\psi}$ can be evaluated using automatic differentiation and the loss can be minimized by gradient based methods. The computation of the expectation value in eq. 1 over the high-dimensional space $\mathbb{R}^{3 \times n_{\text{el}}}$ is done using Monte Carlo integration by sampling electron coordinates $\boldsymbol{r}$ distributed according to $\psi_\theta^2$ using the Metropolis-Hastings [8] algorithm. A thorough discussion of deep-learning-based variational Monte Carlo (DL-VMC) can be found in [9].

**Related work**  Two major neural network architectures and their extensions have emerged throughout literature: PauliNet [9] and FermiNet [10]. PauliNet puts emphasis on maximizing physical prior knowledge, by focusing on the the envelope function. They use the output of CASSCF (Complete Active Space Self Consistent Field, a sophisticated conventional quantum-chemistry method) as $\Omega$ and use a relatively small ($\sim$ 100k weights) neural network for $\Lambda$. FermiNet on the other hand uses a simple exponential function as envelope $\Omega$ and uses a large ($\sim$ 700k weights) neural network for $\Lambda$. Both approaches have been applied with great success to many different systems and properties, such as energies of individual molecules [4, 10, 9], ionization energies [10], potential energy surfaces [11, 12], forces [11], excited states [13], model-systems for solids [14, 15] and actual solids [16]. Several approaches have been proposed to increase accuracy or decrease computational cost, most notably architecture simplifications [17], alternative antisymmetric layers [18], effective core potentials [19] and Diffusion Monte Carlo (DMC) [20, 21]. FermiNet commonly reaches lower (i.e. more accurate) energies than PauliNet[10], but PauliNet has been observed to converge faster [12]. It has been proposed [9] that combining the embedding of FermiNet and the physical prior knowledge of PauliNet could lead to a superior architecture.

**Our contribution**  In this work we present the counter-intuitive observation that the opposite approach might be more fruitful. By combining a PauliNet-like neural network embedding with the envelopes of FermiNet and adding several improvements to the embedding, input features, and initialization of parameter (Sec. 2), we obtain the currently best neural network architecture for the numerical solution of the electronic Schrödinger equation. Combining our new architecture with VMC we establish a new benchmark by calculating the most accurate variational ground state energies ever published for a number of different atoms and molecules - both when comparing to deep-learning-based methods, as well as when comparing to classical methods (Sec. 3). Across systems we reduce energy errors by 40-100% and achieve these results with 3-4x fewer optimization epochs compared to FermiNet. In Sec. 4 we systematically break down which changes cause these improvements. We hypothesize that including too much physical prior knowledge can actually hinder optimization and thus deteriorate accuracy – we provide ample experimental evidence in Sec. 5.

## 2  Improved approach

Similar to FermiNet, our architecture expresses $\Lambda_{ki}^{d}$ as a linear combination of high-dimensional electron embeddings $\boldsymbol{h}_i^L$, and the envelopes $\Omega_k^{d\alpha_i}$ as a sum of exponential functions

$$\Lambda_{ki}^{d}(\boldsymbol{r}) = W_k^{d\alpha_i} \boldsymbol{h}_i^L \qquad \Omega_k^{d\alpha_i}(\boldsymbol{r}_i) = \sum_{I=1}^{n_{\text{nuc}}} \pi_{kI}^{d\alpha_i} \exp(-\omega_{kI}^{d\alpha_i}|\boldsymbol{\rho}_{iI}|), \qquad (3)$$

where $W_k^{d\alpha_i}, \pi_{kI}^{d\alpha_i}, \omega_{kI}^{d\alpha_i}$ are trainable parameters and we enforce $\omega_{kI}^{d\alpha_i} \geq 0$. We compute these embeddings $\boldsymbol{h}_i^L$ by first transforming the inputs $\boldsymbol{R}_I, \boldsymbol{r}_i$ into feature vectors

$$\boldsymbol{h}_i^0 = \left[ |\boldsymbol{\rho}_{iI}|, \tilde{\boldsymbol{\rho}}_{iI} \right]_{I \in \{1,\dots,n_{\text{nuc}}\}} \qquad \boldsymbol{v}_{iI}^0 = \left[ |\boldsymbol{\rho}_{iI}|, \tilde{\boldsymbol{\rho}}_{iI} \right] \qquad \boldsymbol{g}_{ij}^0 = |\boldsymbol{r}_{ij}|$$

where $[\cdot]$ denotes the concatenation operation and then applying $L$ iterations of an embedding network (Fig. 2a). The local difference vectors $\tilde{\boldsymbol{\rho}}_{iI}$ are obtained by applying rotation matrices onto $\boldsymbol{\rho}_{iI}$ as described in Sec. 2.2.

### 2.1  Convolutional layers in embedding

Our embedding network uses four residual neural network streams (Fig. 2b): A primary one-electron stream that embeds a single electron, and three auxiliary streams modelling the two-particle-interactions (electrons with same spins, electrons with different spin, and electron-ion).

$$\boldsymbol{h}_i^{l+1} = \boldsymbol{A}_{\text{one}}^l(\boldsymbol{f}_i^l) + \boldsymbol{h}_i^l \qquad \boldsymbol{g}_{ij}^{l+1} = \boldsymbol{A}_{\sigma_{ij}}^l(\boldsymbol{g}_{ij}^l) + \boldsymbol{g}_{ij}^l \qquad \boldsymbol{v}_{iI}^{l+1} = \boldsymbol{A}_{\text{nuc}}^l(\boldsymbol{v}_{iI}^l) + \boldsymbol{v}_{iI}^l \qquad (4)$$

Here $l$ denotes the embedding iteration, $\boldsymbol{A}^l$ denote fully connected neural networks, and $\boldsymbol{g}_{ij}^l, \boldsymbol{v}_{iI}^l$ denote electron-electron- and electron-nucleus-embeddings. We use $\sigma_{ij} =$ 'same' for same-spin

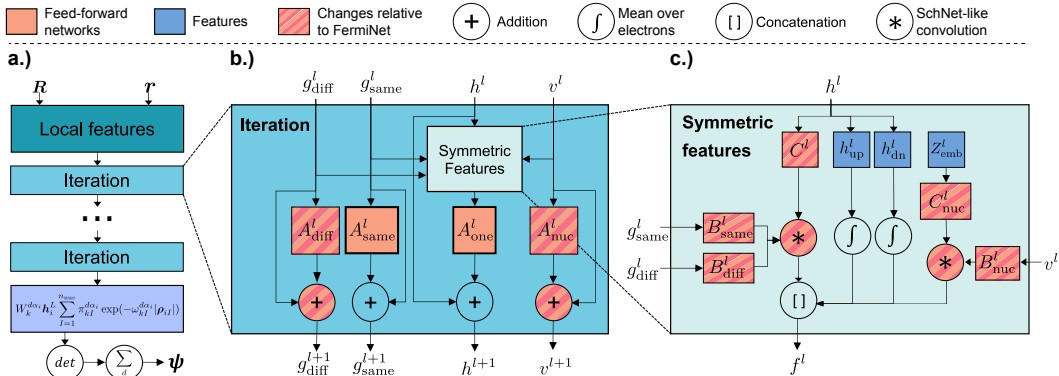

Figure 2: Our architecture: a) High-level overview b) One single embedding iteration c) Sub-block of assembling symmetric features

pairs of electrons and $\sigma_{ij} =$ 'diff' for pairs of electrons with different spin. Similar to FermiNet, in each iteration we assemble the input $\boldsymbol{f}_i^l$ to the primary stream from the auxiliary streams (Fig. 2c):

$$\boldsymbol{f}_i^l = \left[ \boldsymbol{h}_i^l, \quad \frac{1}{n_\uparrow} \sum_{j=1}^{n_\uparrow} \boldsymbol{h}_j^l, \quad \frac{1}{n_\downarrow} \sum_{j=1+n_\uparrow}^{n_{el}} \boldsymbol{h}_j^l, \quad \boldsymbol{s}_i^{l,\text{el}}, \quad \boldsymbol{s}_i^{l,\text{nuc}} \right]. \tag{5}$$

Inspired by the success of SchNet [22] and the efficiency of the PauliNet embedding, we use the sum of element-wise multiplication ($\odot$), effectively forming a convolution, to aggregate the auxiliary two-particle streams:

$$\boldsymbol{s}_i^{l,\text{el}} = \sum_{j=1}^{n_{el}} \boldsymbol{B}_{\sigma_{ij}}^l \left( \boldsymbol{g}_{ij}^l \right) \odot \boldsymbol{C}_{\sigma_{ij}}^l \left( \boldsymbol{h}_j^l \right) \qquad \boldsymbol{s}_i^{l,\text{nuc}} = \sum_{I=1}^{n_{ion}} \boldsymbol{B}_{\text{nuc}}^l \left( \boldsymbol{v}_{iI}^l \right) \odot \boldsymbol{C}_{\text{nuc}}^l \left( Z_I^{\text{emb}} \right) \tag{6}$$

Eq. 5 and 6 form the core of the architecture and are the key difference between FermiNet, PauliNet and our architecture. The PauliNet architecture emphasizes two-particle interactions and essentially only uses convolutions as input features: $\boldsymbol{f}_i^l = [\boldsymbol{s}_i^{l,\text{el}}, \boldsymbol{s}_i^{l,\text{nuc}}]$. In addition to not using the $\boldsymbol{h}_i$ as input features, PauliNet also limits its effective depth by making the convolutional kernels $\boldsymbol{B}$ functions of the electron-electron distances $|\boldsymbol{r}_{ij}|$ instead of the high-dimensional embedded representations $\boldsymbol{g}_{ij}$. The FermiNet architecture on the other hand emphasizes the one-electron stream and only uses sums over $\boldsymbol{g}_{ij}$ as input features, essentially corresponding to $\boldsymbol{B}^l = \text{Id}$, $\boldsymbol{C}(\cdot) = \mathbf{1}$. Furthermore FermiNet does not contain an explicit stream for electron-nucleus interactions. Since our architecture adequately models both the one-electron-embedding as well as the two-particle-interactions, we expect our architecture to be more expressive than either predecessor, as demonstrated in Sec. 4.

## 2.2 Local, invariant input features

The first stage of any VMC wavefunction model is typically the computation of suitable input features from the raw electron coordinates $\boldsymbol{r}$ and nuclear coordinates $\{\boldsymbol{R}_I\}$. While the subsequent embedding stage could in principle take the raw coordinates, appropriate features allow to explicitly enforce symmetries and improve the model's transferability and accuracy. Input features should have three properties: First, they should be sufficiently expressive to encode any physical wavefunction. Second, the features should be invariant under geometric transformations. Third, the features should primarily depend on the local environment of a particle, i.e. similar local geometries should generate similar local features, mostly independent of changes to the geometry far from the particle in question. Published architectures have so far not been able to address all three points: PauliNet [10] uses only distances as input features, making them invariant and local, but not sufficiently expressive, as demonstrated by [12]. FermiNet [10] uses raw distances and differences, making the inputs expressive and local, but not invariant under rotations. PESNet [12] proposes a global coordinate system along the principle axes of a molecule, making the inputs invariant and sufficiently expressive, but not local.

We propose using local coordinate systems centered on every nucleus and evaluating the electron-nuclei differences in these local coordinate systems. Effectively this amounts to applying a rotation

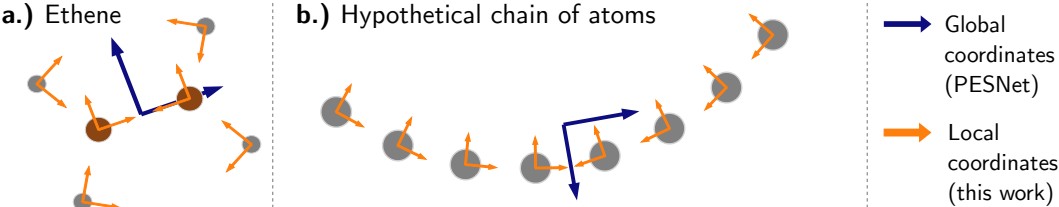

Figure 3: Visualization of resulting coordinate systems for 2 example molecules: a) Ethene b) A hypothetical bent chain of atoms.

matrix $U_J$ to the raw electron-nucleus differences: $\tilde{\boldsymbol{\rho}}_{iJ} = U_J \boldsymbol{\rho}_{iJ}$. As opposed to [12] where $U$ is constant for all nuclei, in our approach $U_J$ can be different for every nucleus. These electron-nucleus differences $\tilde{\boldsymbol{\rho}}_{iJ}$ are invariant under rotations, contain all the information contained in raw Cartesian coordinates and depend primarily on the local environment of an atom. To compute the 3x3 rotation matrices $U_J$, we first run a single Hartree-Fock calculation using a minimal basis set to obtain a density matrix $D$. For each atom $J$ we choose the 3x3 block of the density matrix which corresponds to the 3 p-orbitals of the atom $J$ and compute its eigenvectors $U_J = \mathrm{eig}(D_J)$. To make our coordinate system unique we sort the eigenvectors of $D_J$ by their corresponding eigenvalues. If the eigenvalues are degenerate, we pick the rotation of the eigenvectors that maximizes the overlap with the coordinate-axes of the previous nucleus. Fig. 3 shows the resulting coordinate systems spanned by $U_J$. Note that the local coordinate system generally depicts more physically meaningful directions such as "along the chain". We find that these local coordinates slightly increase the accuracy for single geometries, but more importantly we expect the wavefunction to generalize better across different molecules or geometries. This should improve the accuracy of approaches that attempt to learn wavefunctions for multiple geometries at once [11, 12].

### 2.3 Initialization of orbital envelope weights

When considering a single atom, the entries of the wavefunction determinant have essentially the form

$$\Lambda_{ki}(\boldsymbol{r}) \exp\left(-\omega_k |\boldsymbol{\rho}_{iI}|\right).$$

In [10], the exponential envelope was purely motivated by the boundary condition that the orbitals must decay to zero, and initialization with $\omega_k = 1$ was proposed. However, when comparing this ansatz to analytical solutions, an interesting parallel can be found: Analytical solutions to the Schrödinger equation for atoms with a single electron – the only systems that have analytical solutions – are of the form

$$\widetilde{\Lambda}_k(\boldsymbol{\rho}_{iI}) \exp\left(-\frac{Z}{n_k}|\boldsymbol{\rho}_{iI}|\right),$$

where $\widetilde{\Lambda}_k(\boldsymbol{\rho}_{iI})$ is a product of a Laguerre polynomial and a spherical harmonic, and $n_k \in \mathbb{N}^+$ is known as the principal quantum number. This suggests $\omega_k \approx Z/n_k$, which we also find when analyzing the weights of a fully trained wavefunction. When initializing with $\omega_k = Z/n_k$ instead of $\omega_k = 1$, we observe faster convergence, lower final energies, and lower variance of the energies (Sec. 4). The effect is most notable for molecules containing nuclei with large $Z$, where $Z/n_k \gg 1$.

### 2.4 Improved hyperparameters

Beyond the improved neural network architecture we fine-tuned the hyperparameters to reduce the number of optimization steps required for convergence. Starting from the hyperparameters proposed by [10], we increased the norm constrain by 3x for the second-order optimizer KFAC [23, 24], decreased learning rate by 0.5x, and decreased learning rate decay time by 0.4x. We observe that these changes stabilize the optimization and enable usage of 50% fewer Monte Carlo walkers, which results in ∼2x faster optimization and reduced memory allocation. A complete set of hyperparameters can be found in appendix B.

# 3   Results of improved approach

We evaluated the accuracy of our approach by comparing our computed energies against the most accurate references available in the literature. Fig. 4 compares our energies against variational methods – for which lower energies are guaranteed to be more accurate – as well as non-variational high-accuracy methods. We find that across many different systems (small and large atoms, molecules at equilibrium geometry, molecules in transition states), our approach yields substantially lower – and thus more accurate – energies than previous variational results. Across all tested systems, we outperform almost all existing variational methods, both deep-learning-based methods as well as classical ones. When comparing to high-accuracy FermiNet VMC calculations, we not only reach substantially lower energies, but also do so using 3-4x fewer training steps, with each step being  40% faster (cf. appendix C). Comparing to a concurrently published Diffusion Monte Carlo approach, which used ∼10x more computational resources, we achieve similar or better accuracy for molecules like $N_2$ and cyclobutadiene and slightly lower accuracy for benzene. Non-variational methods (e.g. CCSD(T)) yield slightly lower energies than our calculations for some molecules, but since those methods do not provide upper bounds or uncertainty guarantees they do not provide a ground-truth. For many applications not only absolute energies are important, but energy differences between

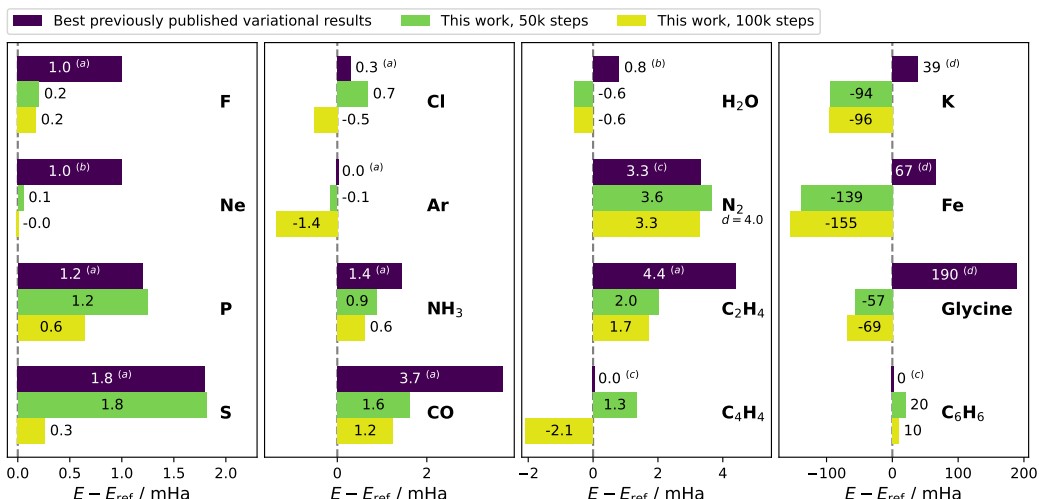

Figure 4: Energies relative to the previously known best estimate, (lower is better). Blue bars depict best published variational energies, footnotes mark the method: a: FermiNet VMC [10, 17], b: Conventional DMC [25, 26, 27], c: FermiNet DMC [21], d: MRCI-F12. A table of absolute energies and methods for $E_{\text{ref}}$ can be found in appendix A. Note that $E_{\text{ref}}$ is not necessary variational and thus may underestimate the true energy.

different molecules or geometries are of interest, for example to determine the energy required to break a chemical bond. A particularly difficult challenge is the dissociation of the $N_2$ molecule, i.e. the energy of an $N_2$ molecule at different bond lengths (inset Fig. 5). Even methods that are generally regarded as highly accurate, such as CCSD(T), predict energies that deviate far from experimental data at bond-lengths from 2.5 - 4 bohr. Fig. 5 depicts this deviation between experimental and computed energy for our method and the best available reference calculations. We find that our results are closer to the experimental absolute energies than all previous work, and are similar to concurrently published FermiNet-DMC results which require 5-10x more computational resources. When comparing relative energies, our approach outperforms all other deep-learning-based methods and CCSD(T), and is only beaten by the highly specialized r12-MR-ACPF method [28]. Similar to absolute energies, we also find that our relative energies converge substantially faster than for other deep-learning-based methods, with relative energies being almost fully converged after 50k epochs.

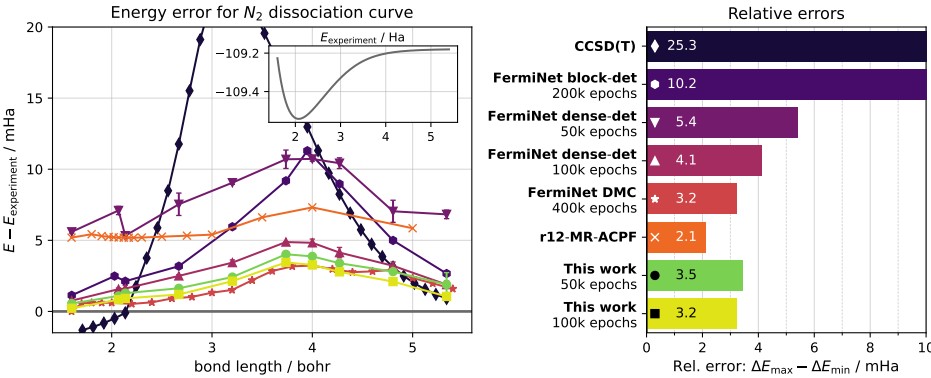

Figure 5: Comparison of energy error $E$ - $E_{\text{experiment}}$ for the dissociation of the $N_2$ molecule across various methods. Errorbars corresponds to the standard deviation wrt. two different seeds. Errorbars for our work are too small to be visible ($\sim$ 0.1 mHa). Results for $E_{\text{experiment}}$ can be found in [29], FermiNet block-det & CCSD(T) in [10], FermiNet DMC in [21] and r12-MR-ACPF in [28].

## 4 Ablation study

To investigate which specific changes lead to the observed improvements in accuracy, we start from the improved FermiNet architecture proposed in [17] and incrementally add improvements in the following order: First, we use dense $n_{\text{el}} \times n_{\text{el}}$ determinants introduced by the FermiNet authors [10, 17] in their GitHub repository and described in [18] instead of block-diagonal determinants. This generalization increases computational cost and parameter count (cf. appendix C) but has been found to better describe the wavefunction's nodal surface and thus increase expressiveness of the ansatz. Second, we change hyperparameters as described in Sec. 2.4, which increases throughput by $\sim$ 2x. Third, we augment the electron embedding using our new SchNet-like neural network architecture described in Sec. 2.1. This leads to a moderate increase in parameter count and computational cost. Fourth, we switch to local, invariant input features as described in Sec. 2.2 and remove the electron-electron difference vectors $\boldsymbol{r}_{ij}$ as inputs. Lastly we switch to initializing $\omega_{kI}^d = {}^Z/n_k$ as described in Sec. 2.3, resulting in our proposed final method. We note that the accuracy gains of these changes are not fully independent of each other and the relative attribution depends on the order in which they are applied: Earlier changes will generally generate larger energy improvements compared to later changes. At each step we compute total energies for three different molecules: ethene, $N_2$ at the challenging bond-length of 4.0 bohr, and the K-atom. Fig. 6 depicts the energy of our implementation of FermiNet, and the energy change caused by each subsequent improvement. Each experiment was repeated two times with different RNG seeds (appendix D), the errorbars depict the spread in energy. Overall we find that all our changes combined yield a $\sim$3-20x improvement in the energy error. For ethene, the dominant contribution (3.7 mHa) comes from improved hyperparameters, which lead to the results being mostly converged after 50k epochs vs. the original settings which require 200k epochs for convergence. Using a lower learning rate in combination with a larger gradient-norm-constraint ensures that more optimization steps are taken according to curvature estimated by KFAC and fewer steps are clipped by the gradient-norm-constraint. Architectural improvements (embedding and input features) lower the energy error by additional 1.4 mHa. Because our embedding is a strict generalization of both FermiNet and PauliNet, our ansatz is more expressive and can therefore reach lower energies than previous ansätze. For $N_2$ it has already been observed that a single dense determinant can outperform models with multiple block-diagonal determinants [18]. We find the corresponding result that 32 dense determinants substantially lower the energy relative to an ansatz with 32 block-diagonal determinants. Comparing $N_2$ to ethene, we observe larger contributions from our architectural improvements and smaller contributions from improved hyperparameters. For the K atom, the overall gains are largest, totalling 60mHa, with substantial accuracy gains from all improvements. Since K has a much larger nuclear charge (Z=19) than the constituents of ethene (Z=1,6) and $N_2$ (Z=7), also the physics-inspired initialization of the envelope parameters yields a substantial contribution. This improved initialization leads to a better initial guess for the wavefunction, which not only reduces the number of required optimization steps, but also leads to more accurate initial sampling.

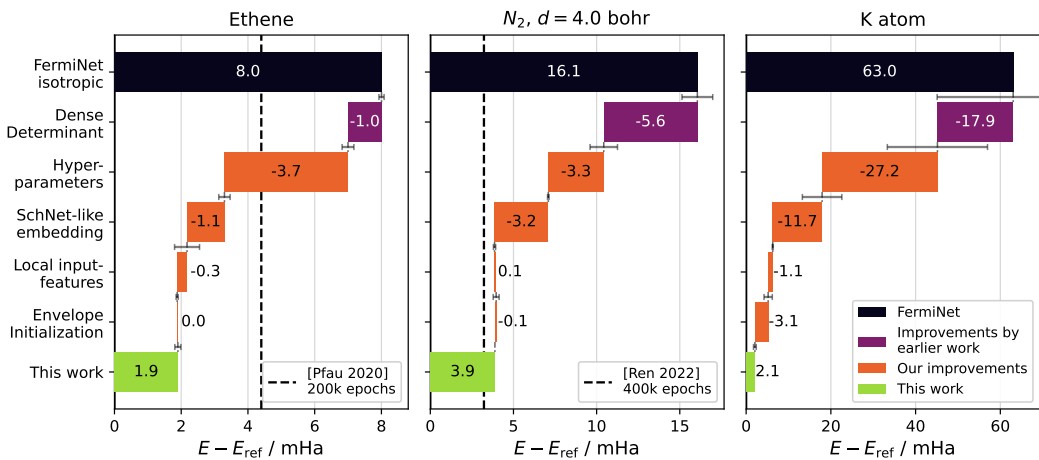

Figure 6: Breakdown of accuracy improvements for three different molecules, each trained for 50k epochs. The black dashed line depicts the best published variational result and as reference we use the best available estimates: CCSD(T) [10] for ethene, experimental data for $N_2$ [29], and our own VMC calculations after 100k epochs for the K atom.

## 5    Incorporating prior knowledge

To further understand the effect of incorporating prior knowledge into neural network architectures for physical problems as the electronic Schrödinger equation, we examined two distinct ways of increasing prior information in our model: First, by including a built-in approximate physical model, analogous to PauliNet. Second, by increasing the number of pre-training steps to more closely match a reference wavefunction before starting the optimization.

**Explicitly include CASSCF**    PauliNet maximizes the physical prior information by computing the envelopes $\Omega$ with CASSCF, a sophisticated conventional method, and explicitly enforcing the Kato cusp conditions [30]. Starting from our proposed architecture, we modified our approach step-by-step until we arrived at a PauliNet-like architecture. Fig. 7a shows the energies of an $NH_3$ molecule trained for 50k epochs at each step.

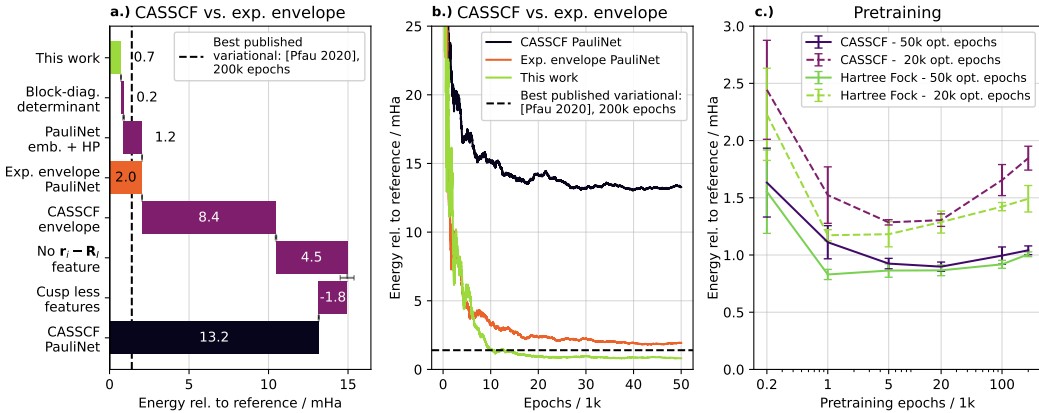

Figure 7: Effect of prior knowledge for $NH_3$: a) Change in accuracy after 50k optimization epochs when transitioning from our approach to a PauliNet-like architecture, by increasing prior knowledge. b) Energy error for 3 architectures over number of training epochs. c) Mean and standard deviation of the energy error as a function of pre-training steps. Pre-training used CASSCF or HF as a reference, and the energy was evaluated after 20k or 50k subsequent variational optimization epochs.

First, we switch from dense determinants to block-diagonal determinants as used by the original PauliNet, leading to small loss in accuracy. Second, we exchange our embedding for the PauliNet-like embedding using the hyperparameters proposed in [11], leading to a substantial loss in accuracy, presumably caused by a loss in expressiveness. Next, we replace the simple exponential envelopes by the more physically inspired CASSCF-envelopes, causing a large loss in accuracy. We then remove the vector $r_i - R_I$ as input feature (keeping only its absolute value) as done in the original PauliNet architecture [9]. This again deteriorates accuracy, presumably due to enforcing rotational invariance which is too restrictive of a symmetry class as pointed out by [12]. Lastly we replace the electron-electron distances $|r_{ij}|$ (which are not smooth at $r_{ij} = 0$ and thus lead to cusps) by smooth, cuspless radial basis functions as input feature and add an explicit term to enforce the electron-electron cusp condition. Since the CASSCF-envelopes are designed to fulfill the electron-ion cusp condition, this change leads to an enforced cusp condition, slightly improving the energy accuracy. The largest loss in accuracy throughout these changes is caused by introducing the CASSCF-envelopes, suggesting that they introduce a strong bias of the wavefunction that cannot be easily overcome during training. Fig. 7b shows that architectures using exponential envelopes converge to lower absolute energies compared to the CASSCF-based PauliNet and outperform PauliNet already after $\sim$5000 epochs.

**Increase pre-training accuracy**    Before starting the unsupervised variational optimization of the wavefunction, we run a short supervised pre-training of the wavefunction to roughly match a given reference wavefunction. This is computationally inexpensive because it only requires evaluation of $\psi_\theta$ and a back-propagation step to update the neural network weights $\theta$ but not the second derivative of the Hamiltonian. If the reference method yields a decent approximation to the true wavefunction, this pre-training significantly speeds-up optimization and avoids unstable parameter regimes [10]. To incorporate more prior knowledge, one could either use a more sophisticated reference method (e.g. CASSCF instead of HF) or increase the number of pre-training steps. In Fig. 7c we pre-trained the wavefunction with a block diagonal determinant for the $NH_3$ molecule using a CASSCF and Hartree-Fock reference. We increased pre-training iteration steps and evaluated the energy after subsequent 20k and 50k variational optimization epochs, each run was repeated with five different seeds. Increasing the number of pre-training steps initially increases accuracy – since it provides a better starting point for the subsequent variational optimization – but when increasing pre-training beyond 20k steps, accuracy deteriorates for both methods. Surprisingly, we observe a drop in accuracy when using CASSCF as a reference method compared to the simpler Hartree-Fock method. This effect is even more pronounced when increasing the number of pre-training steps. It suggests that excessive pre-training introduces a bias that is hard to overcome during variational optimization, similarly to a built-in reference method.

# 6    Discussion and Limitations

**Discussion**    We find that our approach yields substantially more accurate absolute energies than all previous work – both classical as well as deep-learning-based – and that we reach these accuracies 4-6x faster than the next best method (FermiNet). Especially for larger systems, such as 4th row atoms or the amino acid glycine, we outperform conventional "gold-standard" methods like MRCI-F12(Q) by $\sim$100 mHa. This corroborates the fact that deep-learning-based methods are emerging as a new gold-standard in computational chemistry and showcases the immense potential of machine-learning-based methods in the natural sciences. A concurrent work [21] was able to achieve similar accuracies by applying Diffusion Monte Carlo (DMC) on top of a FermiNet VMC calculation, highlighting the potential of deep-learning Monte Carlo methods. However, [21] required $\sim$10x more computational resources and their VMC results – already by themselves 8x more expensive then our calculations – are consistently inferior to our results. This showcases a promising route towards further improvements by using our substantially cheaper and more accurate VMC results as a starting point for a DMC calculation.

Regarding the question of how much physics to include in the model, we find varying results. For exact physical constraints, such as symmetries or the cusp conditions, inclusion in the model generally appears to be helpful. However for prior knowledge from existing approximate solutions (such as CASSCF) the situation is more subtle. On the one hand, soft physical guidance such as short supervised pre-training or physics-inspired weight initialization accelerates optimization. On the other hand, we show empirically that increasing physical prior knowledge, e.g. by incorporating

CASSCF or extensive supervised pre-training, does not necessarily increase accuracy, but can in fact introduce detrimental biases that are hard to overcome during wavefunction optimization.

**Limitations and outlook** Despite the proposed improvements and favorable scaling of the method, computation of energies for large molecules still takes days of GPU-time on current hardware. While the same holds true for conventional high-accuracy approaches, substantial speed-ups are still required to make DL-VMC more accessible for practitioners. Additionally, when increasing the nuclear charges, the wavefunction becomes increasingly localised, which leads to a reduction in average Monte Carlo stepsize and potentially correlated samples. We circumvent this effect for 4th row atoms by increasing the number of intermediate Monte Carlo steps, but further research into Monte Carlo sampling methods [31, 32] is required to fully address this issue. Despite our improvements for the accuracy of energy differences between different molecules or geometries, DL-VMC is still outperformed by other, computationally cheaper methods in some cases. Initial research into the regularity of the wavefunction across different molecules [11, 12] provides a promising route to improvements. We note in passing that thanks to the local coordinate input features, our architecture fulfills the required rotational invariance required for these approaches.

# 7 Code availability

The code alongside a detailed documentation is available as part of the DeepErwin code package on the Python Package Index (PyPI) and github (`https://github.com/mdsunivie/deeperwin`) under the MIT license.

# 8 Acknowledgements

We gratefully acknowledge financial support from the following grants: Austrian Science Fund FWF Project I 3403 (P.G.), WWTF-ICT19-041 (L.G.). The computational results have been achieved using the Vienna Scientific Cluster (VSC). The funders had no role in study design, data collection and analysis, decision to publish or preparation of the manuscript. Additionally, we thank Nicholas Gao for providing his results and data and Rafael Reisenhofer for providing valuable feedback to the manuscript.

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
