# Supplementary Information - Gold-standard solutions to the Schrödinger equation using deep learning: How much physics do we need?

## A Absolute energies

Table 1: Ground-state energy in Ha for various methods grouped into best variational energy, lowest overall and this work. The lowest variational energy is highlighted with bold letters and the lowest overall energy by an underscore. As reference we included methods with the best energy approximation found in the literature and if not found elsewhere own computations based on a MRCI method were performed. a: FermiNet VMC [1, 2], b: FermiNet DMC [3], c: Diffusion Monte Carlo [4, 5, 6], d: MRCI-F12, e: CCSD(T)/ CBS [1], f: CAS + experimental corrections [7], g: MRCI-F12(Q), h: experiment [8].

| Category | System | Lowest Variational | Lowest All | This work 50k epochs | This work 100k epochs |
|---|---|---|---|---|---|
| 2nd row atoms | O | $-75.0666$ [a] | $-\underline{75.0673}$ [f] | $-75.0669$ | $\mathbf{-75.0670}$ |
| | F | $-99.7329$ [a] | $-\underline{99.7339}$ [f] | $\mathbf{-99.7337}$ | $\mathbf{-99.7337}$ |
| | Ne | $-128.9366$ [c] | $-\underline{128.9376}$ [f] | $-128.9375$ | $\mathbf{-128.9376}$ |
| 3rd row atoms | P | $-341.2578$ [a] | $-\underline{341.259}$ [f] | $-341.2578$ | $\mathbf{-341.2584}$ |
| | S | $-398.1082$ [a] | $-\underline{398.110}$ [f] | $-398.1082$ | $\mathbf{-\underline{398.1097}}$ |
| | Cl | $-460.1477$ [a] | $-460.148$ [f] | $-460.1473$ | $\mathbf{-\underline{460.1485}}$ |
| | Ar | $-527.5405$ [a] | $-527.5405$ [a] | $-527.5406$ | $\mathbf{-\underline{527.5419}}$ |
| 4th row atoms | K | $-599.7847$ [d] | $-599.8238$ [g] | $-599.9178$ | $\mathbf{-\underline{599.9195}}$ |
| | Fe | $-1263.4281$ [d] | $-1263.4947$ [g] | $-1263.633$ | $\mathbf{-\underline{1263.650}}$ |
| Small molecules | $N_2, d = 2.068$ | $\mathbf{-109.5416}$ [b] | $-\underline{109.5425}$ [e] | $-109.5408$ | $-109.5414$ |
| | Ethene | $-78.5844$ [a] | $-\underline{78.5888}$ [e] | $-78.5868$ | $\mathbf{-78.5871}$ |
| | $H_2O$ | $-76.4368$ [c] | $-76.4376$ [h] | $-76.4382$ | $\mathbf{-\underline{76.4382}}$ |
| | CO | $-113.3218$ [a] | $-\underline{113.3255}$ [e] | $-113.3239$ | $\mathbf{-113.3242}$ |
| | $NH_3$ | $-56.5630$ [a] | $-\underline{56.5644}$ [e] | $-56.5635$ | $\mathbf{-56.5638}$ |
| Larger molecules | Cyclobutadiene | $-154.6770$ [b] | $-154.67700$ [b] | $-154.6757$ | $\mathbf{-\underline{154.6791}}$ |
| | Benzene | $\mathbf{-232.2370}$ [b] | $\mathbf{-232.2370}$ [b] | $-232.2168$ | $-232.2267$ |
| | Glycine | $-284.1741$ [d] | $-284.3639$ [g] | $-284.4201$ | $\mathbf{-\underline{284.4328}}$ |

## B Computational settings

**Neural network** The main hyperparameters used in this work can be found in Tab. 2 with minor changes for larger systems. For the third and fourth row atoms we increased as in [2] the single electron stream to $512$. For K and Fe we choose a higher damping for the second-order optimizer KFAC with $4 \times 10^{-3}$ to stabilize the optimization. Additionally, we observed that with increasing

nuclear charge $Z$ the MCMC stepsize tends to decrease because of a high probability peak around the core resulting in two distinct length scales for the core and valence electrons. To circumvent high correlation between samples we increased the number of decorrelation steps to 60 for Fe. We observed for larger systems that 1000 pretraining steps is not sufficiently close to a Hartree-Fock solution and therefore increased for glycine, benzene, cyclobutadiene, K and Fe the number of pretraining steps to 4000. For glycine and benzene the batch size and number of MCMC walkers was decreased to 1700 allowing us to use a single NVIDIA A100 GPU and for glyzine, benzene and cyclobutadiene no envelope initialization at the beginning of optimization was performed. All computations were computed on either a single NVIDIA A100 or A40 GPU. We use separate 10k evaluation steps in which we do not update the neural network parameters to compute all final energies, to ensure fully unbiased energies.

Table 2: Hyperparameter settings used in this work

| | | |
|---|---|---|
| **Orbitals** | # determinants | 32 |
| | Pretraining basis set | 6-311G |
| | Pretraining steps | 1000 |
| **Embedding** | # determinants $n_{\mathrm{det}}$ | 32 |
| | # hidden layers $A_{\mathrm{one}}, A_{\sigma_{i,j}}, A_{\mathrm{ion}},$ $B_{\mathrm{ion}}, B_{\sigma_{i,j}}, C_{\mathrm{ion}}, C_{\sigma_{i,j}}$ | 0 |
| | # neurons per layer $A_{\mathrm{one}}$ | 256 |
| | # neurons per layer $A_{\sigma_{i,j}}, A_{\mathrm{ion}},$ $B_{\mathrm{ion}}, B_{\sigma_{i,j}}, C_{\mathrm{ion}}, C_{\sigma_{i,j}}$ | 32 |
| | # iterations $L$ | 4 |
| | activation function | tanh |
| **Markov Chain Monte Carlo** | # walkers | 2048 |
| | # decorrelation steps | 20 |
| | Target acceptance probability | 50% |
| **Optimization** | Optimizer | KFAC |
| | Damping | $1 \times 10^{-3}$ |
| | Norm constraint | $3 \times 10^{-3}$ |
| | Batch size | 2048 |
| | Initial learning rate $\mathrm{lr}_0$ | $5 \times 10^{-5}$ |
| | Learning rate decay | $\mathrm{lr}(t) = \mathrm{lr}_0(1 + t/6000)^{-1}$ |
| | Optimization steps | 50,000 - 100,000 |

**Reference calculation with MRCI**   We carried out reference calculations for several small molecules (see Table 3) and atoms (see Table 4) using the MOLPRO package [9, 10]. On the one hand, as single-reference method, we tested coupled cluster with single and double excitations including perturbative triple excitations in its explicitly correlated F12 formulation (CCSD(T)-F12) [11], which requires a restricted Hartree-Fock (HF) calculation in the employed implementation. On the other hand, as multi-reference methods, we used multi-reference configuration interaction in its explicitly correlated F12 formulation (MRCI-F12), which necessitates a complete active space self-consistent field (CASSCF) calculation. We also carried out perturbative Davidson corrections on the MRCI-F12 values (abbreviated as MRCI-F12(Q)). Among all of these methods, HF, CASSCF (as used here in its state-specific variant) and MRCI-F12 are variational. In contrast, CCSD(T)-F12 and MRCI-F12(Q) contain perturbative contributions to the ground-state energy and, thus, do not obey the variational principle.

The active spaces for the molecules and atoms is denoted here in the format [multiplicity](number of electrons, number of orbitals) and was chosen as: $H_2O$ [1](8,6), cyclobutadiene [1](4,4), benzene [1](6,6), glycine [1](6,4), K [2](1,6), Fe [5](8,6). For the molecules, the correlation-consistent quadruple-zeta basis set cc-pVQZ-F12 [12] was employed together with the recommnended parameter GEM_BETA = 1.5 $a_0^{-1}$ [13] (which lead to convergence problems for the benzene multi-reference calculation). In order to judge the basis set convergence, also calculations with the smaller triple-zeta basis set cc-pVTZ-F12 was carried out together with GEM_BETA = 1.4 $a_0^{-1}$. These basis sets were not

available for the atoms and the def2-QZVPP basis set together with the corresponding density-fitting def2-QZVPP-JKfit basis were employed instead [14].

Table 3: Ground-state energies of selected small molecules computed with various combinations of methods and basis sets, as indicated.

| Molecule | Basis set | $E_{HF}$ | $E_{CASSCF}$ | $E_{CCSD(T)-F12}$ | $E_{MRCI-F12}$ | $E_{MRCI-F12(Q)}$ |
|---|---|---|---|---|---|---|
| $H_2O$ | cc-pVQZ-F12 | -76.0673 | -76.1208 | -76.3766 | -76.4210 | -76.4352 |
| | cc-pVTZ-F12 | -76.0669 | -76.1189 | -76.3752 | -76.4183 | -76.4324 |
| Cyclo-butadiene | cc-pVQZ-F12 | -153.7060 | -153.7598 | -154.4607 | -154.5288 | -154.6380 |
| | cc-pVTZ-F12 | -153.7054 | -153.7574 | -154.4597 | -154.5250 | -154.6340 |
| Benzene | cc-pVQZ-F12 | -230.7970 | – | -231.9115 | – | – |
| | cc-pVTZ-F12 | -230.7960 | -230.8580 | -231.9099 | -231.9584 | -232.1483 |
| Glycine | cc-pVQZ-F12 | -282.9740 | -282.9902 | -284.1623 | -284.1741 | -284.3639 |
| | cc-pVTZ-F12 | -282.9730 | -282.9847 | -284.1587 | -284.1671 | -284.3570 |

Table 4: Ground-state energies of selected atoms computed with the indicated methods in combination with a def2-QZVPP basis set and the corresponding density-fitting def2-QZVPP-JKfit basis set.

| Atom | $E_{CASSCF}$ | $E_{MRCI-F12}$ | $E_{MRCI-F12(Q)}$ |
|---|---|---|---|
| K | -599.1645 | -599.7847 | -599.8238 |
| Fe | -1262.4459 | -1263.4281 | -1263.4947 |

## C  Computational cost

For a better comparison of the computational cost of our proposed architecture to FermiNet we report in Tab. 5 the runtime per optimization epoch and the number of all trainable parameters for all settings tested in Sec. 4. The majority of the runtime per epoch is due to the second derivative for the kinetic energy of the Hamiltonian. Comparing runtime for the K atom we are $\sim 40\%$ faster per epoch then FermiNet with a block determinant structure and even $\sim 60\%$ faster when using a full determinant. The option of full determinant comes not free of costs and increases substantially the number of total trainable parameters and a slightly more expensive optimization iteration for FermiNet. This effects is compensated by a faster convergence as has been seen in Fig. 6. To enable a full like-for-like comparison we compare all runtimes vs. our own implementation of FermiNet, which achieves the same runtime as the original code. Additionally, we state in Tab. 6 the improvements and speed-ups compared to previously published FermiNet results. Speed-ups are a product of a lower number of training iterations times the faster runtime per optimization epoch (cf. Tab. 5). Energy improvements are obtained by taking the ratio of each methods' energy error with respect to a ground truth solution. In three cases the ground-truth is represented by our computations. We used approximately 50k GPUh in total for this work (10k on A100 GPUs, 40k on A40 GPUs).

Table 5: Comparing the architectures from Fig. 6 with respect to runtime per optimization epoch and number of trainable parameters. All timings were performed using a single NVIDIA A100 GPU.

| System | Architecture Setting | Runtime per opt. epoch | Number of parameters |
|---|---|---|---|
| K atom | FermiNet isotropic (our implementation) | 5.1 s | 2,784,672 |
| | Dense Determinant | 6.0 s | 3,097,184 |
| | Hyperparameters | 3.1 s | 3,097,184 |
| | SchNet-like embedding | 3.8 s | 3,175,492 |
| | Local input features | 3.6 s | 3,175,108 |
| | Envelope initialization (This work) | 3.6 s | 3,175,108 |

Table 6: Energy improvements and speed-ups compared to FermiNet. Energy improvements are computed with respect to the best estimate. For three systems our method improves the best estimate and therefore the best estimate is our calculated energy. Speed ups are computed as factor of faster convergence due to fewer training epochs times a 40% (5.1s/3.6s) faster single epoch (cf. Tab. 5).

| System | FermiNet epochs | This work, 50k epochs | | This work, 100k epochs | |
|---|---|---|---|---|---|
| | | accuracy gain | speed-up | accuracy gain | speed-up |
| F | 200,000 | 61% | 6x | 63% | 3x |
| Ne | 200,000 | 94% | 6x | 100% | 3x |
| P | 350,000 | -4% | 10x | 46% | 5x |
| S | 350,000 | 0% | 10x | 85% | 5x |
| Cl | 350,000 | -48% | 10x | 100% | 5x |
| Ar | 350,000 | 11% | 10x | 100% | 5x |
| $NH_3$ | 200,000 | 38% | 6x | 57% | 3x |
| CO | 200,000 | 45% | 6x | 53% | 3x |
| $N_2$ | 200,000 | 52% | 6x | 69% | 3x |
| $C_2H_4$ | 200,000 | 53% | 6x | 61% | 3x |
| $C_4H_4$ | 200,000 | 71% | 6x | 100% | 3x |

## D    Run-to-run variation

Due to the stochastic nature of initialization and Monte Carlo sampling, our results show small run-to-run variations, depending on the initialization of the random number generator. To estimate the extent of the variation we repeated each experiment in the ablation study two times, and depicted mean and standard deviation of these two runs in Fig. 6. To further quantify this uncertainty, we repeated the optimization of an $N_2$ molecule 10 times. We used two different settings (our proposed approach and FermiNet using dense-determinants, before applying our improvements) and optimized for 50k epochs. We find a standard deviation of 0.2 mHa for our approach and 0.8 mHa for the FermiNet approach. We attribute the lower spread of our approach to the fact that our results are already converged after 50k epochs (with the remaining uncertainty primarily caused by Monte Carlo evaluation uncertainty), while the unconverged FermiNet results depend more strongly on the random initialization.

## E    Broader impact

Advancements in computational chemistry may prompt new discoveries in chemistry and biology, which could include positive outcomes such as the development of new drugs or materials. Like every computational chemistry method, our work could hypothetically also be misused for the development of chemical weapons or other potential risks to humanity. As of now, this seems highly unlike due to basic nature of our research.